# Secondary Amenorrhea and Infertility Due to an Inhibin B Producing Granulosa Cell Tumor of the Ovary. A Rare Case Report and Literature Review

**DOI:** 10.3390/medicina57080829

**Published:** 2021-08-17

**Authors:** Corina Gică, Ruxandra-Gabriela Cigăran, Radu Botezatu, Anca Maria Panaitescu, Brîndușa Cimpoca, Gheorghe Peltecu, Nicolae Gică

**Affiliations:** 1Department of Obstetrics and Gynecology, Filantropia Clinical Hospital, 71117 Bucharest, Romania; corina.gica@drd.umfcd.ro (C.G.); radu.botezatu@umfcd.ro (R.B.); anca.panaitescu@umfcd.ro (A.M.P.); brindusa.cimpoca@gmail.com (B.C.); gheorghe.peltecu@umfcd.ro (G.P.); gica.nicolae@umfcd.ro (N.G.); 2Department of Obstetrics and Gynecology, Carol Davila University of Medicine and Pharmacy, 71117 Bucharest, Romania

**Keywords:** fertility-sparing surgery, ovarian cancer, oophorectomy, inhibin B, granulosa cell tumor

## Abstract

Granulosa cell tumor of the ovary (GCT) is a rare ovarian tumor with nonspecific symptoms. Studies reported that GCT are usually secreting estrogens and inhibins, especially inhibin B. It is considered that, in premenopausal women, irregular menses or secondary amenorrhea may be an early symptom of GCT and, in postmenopausal women, the most common manifestation is vaginal bleeding. Additionally, endometrial abnormalities can be associated due to estrogenic secretion. At reproductive age, high levels of inhibin, lead to low levels of FSH and secondary amenorrhea causing infertility. At times, increased levels of LH in women with GCT are observed and the pathogenesis is still unclear. Therefore, inhibin B level can differentiate GCT from other causes of secondary amenorrhea. We report the case of a 26-year-old nulliparous, women who presented in our clinic with secondary infertility lasting longer than 2 years, secondary amenorrhea, polycystic ovarian syndrome, and suspicion of right ovarian endometrioma on CT scan. The ultrasound examination revealed that the right ovary was transformed in an anechoic mass with increased peripheral vascularity having a volume of 10 cm^3^. This patient had high serum levels of inhibin B and LH but normal levels of FSH and estradiol. The preliminary diagnosis of granulosa cell tumor of the ovary was made. After counseling, the informed consent for treatment was obtained and the patient agreed to undergo surgery. An uneventful laparoscopy was performed with right oophorectomy and multiple peritoneal sampling. The histological diagnosis confirmed adult GCT limited to right ovary, with negative peritoneal biopsies (FIGO IA). After surgery the patient recovered fully and had normal menstrual cycles with normal serum levels of hormones. Two months later she conceived spontaneously and had an uneventful pregnancy. In conclusion, for cases with secondary amenorrhea, the evaluation of inhibin B level is essential. Elevated inhibin B level may be a sign for the presence of an unsuspected tumor. With early diagnosis and treatment, the prognosis is generally good and the fertility may be preserved, especially in young patients with GCT.

## 1. Introduction

Granulosa cell tumor of the ovary (GCT) is a relatively rare ovarian tumor with low incidence, representing 5% of all ovarian cancers [1]. Thus, limited data about this condition is available [2]. Regarding the histological classification, there are two subtypes: adult-type and juvenile-type [3]. The adult-type is frequently diagnosed in premenopausal or early postmenopausal period (average age 50–54 years) and the juvenile-type is diagnosed in premenarchal girls and young women. Exceptionally, the adult-type is seen in children or young women [4,5]. These tumors are usually described as large ovarian masses (>10 cm in size), but their size can vary from 3 cm to 24 cm [3].

It is believed that GCT are derived from granulosa cells, somatic cells of the sex cords of the ovary. They are responsible for production of sex steroids (estradiol) and proteins (inhibin) important in folliculogenesis and ovulation [3]. Through feedback mechanism, estradiol increases LH secretion and decreases FSH secretion. Inhibin, as its name shows, inhibits FSH secretion. Studies reported that GCT are usually producing estrogens and inhibins, especially inhibin B. At reproductive age, high levels of inhibins lead to low levels of FSH and secondary amenorrhea, therefore are causing infertility [6,7].

Symptoms are nonspecific, abnormal vaginal bleeding or amenorrhea are the most common ones. A palpable abdominal mass and abdominal pain are frequently seen in patients with GCT [5,8,9]. Additionally, endometrial abnormalities (endometrial hyperplasia or endometrial carcinoma) can be associated in these patients, mainly due to estrogen secretion [5,10].

There are no specific imaging ultrasound (US) criteria for the diagnosis of these tumors and it is sometimes misdiagnosed as endometrial cyst, especially in cases where infertility is associated [1]. Ko SF et al. reported that imagistic patterns of these tumors vary from solid masses to cystic tumors with different aspects (multilocular, hemorrhagic, or fibrotic content) [11].

The standard treatment for GCT is complete surgery consisting of hysterectomy with bilateral salpingo-oophorectomy. Fertility-preserving surgery with unilateral salpingo-oophorectomy with histological staging is an option in young patients with stage IA GCT [12]. Chemotherapy is recommended for patients with advanced stages and recurrent disease. In early stages of GCT, only a selected population should receive adjuvant chemotherapy. These are high risk patients with large or ruptured tumors or masses with high mitotic index) [12]. The prognosis for women with GCT is generally good, especially for those diagnosed in early stages [5].

After removal of tumor, serological levels of hormones return to normal. Serum levels of inhibin B are useful in the follow-up of granulosa cell tumors [13].

## 2. Case Report

We report a case of a 26-year-old nulliparous woman who presented in our clinic with secondary infertility lasting longer than 2 years. She was referred to our clinic with polycystic ovarian syndrome and secondary amenorrhea. She was experiencing menstrual bleeding only after progesterone withdrawal. She had a history of 3 voluntary pregnancy terminations with the same partner and her medical history was otherwise nonsignificant. Among her investigations, the patient had a CT scan raising the suspicion of right ovarian endometrioma. The pelvic ultrasound examination revealed that the right ovary was transformed in an anechoic mass with increased peripheral vascularity, reaching a volume of 10 cm^3^ and no sign of normal peripheral ovarian tissue or ultrasonographic criteria for polycystic ovaries (Figure 1).

The uterine cavity was normal, with linear, suppressed, thin endometrium. Hormonal status was determined and the results were: anti-Mullerian hormone (sAMH) was 1.86 ng/mL, luteinizing hormone (LH) 41.9 mUI/mL, FSH 7 mUI/mL, Estradiol 168.5 pmol/L, Testosterone 1.46 nmol/L, Progesterone 1.21 pmol/L, total HCG 1.4 mUI/mL, HE4 56.53 pmol/L, AFP 1.66 ng/mL, CA 125 20.7 mUI/mL, and Inhibin B 200 pg/mL.

The preliminary diagnosis of granulosa cell tumor of the ovary was made. After counseling, the informed consent for treatment was obtained and the patient agreed to undergo surgery. A laparoscopy was performed and the right ovary was described as a white incapsulated tumor, without extracapsular disease (Figure 2). The left ovary had normal aspect left tube had proximal obstruction and the right tube was normal, with positive methylene-blue test. There was no macroscopic evidence of metastasis or evidence for residual tumor. The decision of right oophorectomy was made and multiple peritoneal biopsies were collected.

The histopathology report confirmed the clinical suspicion of adult GCT limited to right ovary, with all peritoneal biopsies being negative. The oncology staging was FIGO IA and the patient was given the options for in vitro fertilization (IVF) due to left tubal obstruction with higher rates of success, or conservative management regarding a future pregnancy.

For the FIGO IA GCT she was seen by the multidisciplinary team for clinical and imaging monitoring. After surgery the patient recovered fully and had normal menstrual cycles with normal serum levels of hormones. Two months later she conceived spontaneously and had an uneventful pregnancy (Figure 3). The patient consented and agreed with the publication of the information in this case report.

## 3. Discussion and Conclusions

Clinical symptoms for granulosa cell tumors are nonspecific, making the diagnosis difficult and, therefore, delaying the treatment. In young women, secondary amenorrhea usually occurs due to increased levels of inhibin B which subsequently inhibits the secretion of pituitary follicle-stimulating hormone (FSH) [14]. In the presented case, both FSH and estradiol were within normal range, the patient had normal levels of testosterone, and the AMH level was 1.86 ng/mL. On the other hand, the serum levels of inhibin B and LH were high causing anovulation and, therefore, secondary amenorrhea. Thus, it is important to take into consideration the possibility of an ovarian tumor in women of reproductive age with secondary amenorrhea [13]. Several cases of GCT with secondary amenorrhea have been described in the literature (Table 1). Lappöhn et al. reported three cases, in two cases FSH levels were below the normal range, and, in one case, FSH and LH levels were low. As shown in the Table 1, in the majority of cases, FSH levels were below the normal range [13].

Nasu et al. described a case with high level of LH and normal FSH and estradiol levels [15]. Additionally, Ran et al. reported a case with LH elevation and normal FSH level, but low estradiol level [16]. In the presented case, LH level was high and FSH level was normal. Increased level of LH in patients with GCT is rare and its pathogenesis is still unclear. The literature mentions GnRH-like substances from normal granulosa cells may be responsible for LH elevation [16,17]. The LH receptor mutation was, also, incriminated in the increased LH secretion [18]. However, these remarks need to be confirmed with further studies. For these reasons, inhibin B level can differentiate GCT from other causes of secondary amenorrhea. It is considered that, in premenopausal women, irregular menses or secondary amenorrhea may be the initial symptom of a GCT and, in postmenopausal women, the most common manifestation is vaginal bleeding due to the estrogen-producing tumor [19,20].

Additionally, patients with GCT present in 50% of cases endometrial hyperplasia due to the high levels of estrogens, therefore transvaginal ultrasound examination is recommended to assess the endometrial thickness and decide if endometrial biopsy is necessary [3,21]. Endometrial cancer is diagnosed in 5–13% cases [22]. Lee et al. reported endometrial abnormalities (hyperplasia or cancer) associated with GCT in 18 cases out of 69 cases [5]. Additionally, Babarovic et al. described endometrial disorders at 17 patients out of 33 evaluated patients [23].

An early diagnosis means a quicker treatment and, therefore, a better prognosis and the possibility of preserving fertility, especially for young patients with GCT [5].

**Table 1 medicina-57-00829-t001:** Endocrine changes in patients with GCT and secondary amenorrhea—literature review. NA: not applicable.

Authors	Year	Journal	Cases	FSH	LH	Estradiol	Inhibin B
Lappöhn et al. [13]	1989	N Engl J Med	2 cases	low	normal	NA	high
Lappöhn et al. [13]	1989	N Engl J Med	1 case	low	low	NA	high
Sakamoto et al. [24]	1998	Acta Obstet. Gynecol. Scand.	1 case	low	normal	NA	high
Krishnan et al. [25]	2003	Human Reproduction	1 case	low	normal	normal	high
Kurihara et al. [19]	2004	J Obstet Gynaecol Res.	1 case	low	normal	normal	high
Nasu et al. [15]	2007	International J of Clinical Oncology	1 case	normal	high	normal	NA
Agha-Hosseini et al. [6]	2008	Taiwan J Obstet Gynecol	1 case	low	low	normal	high
Tasci et al. [26]	2015	Turkish J of Gynecological Oncology	2 cases	low	normal	normal	high
Ran et al. [16]	2017	J of Ovarian Research	1 case	normal	high	low	NA
Okawa et al. [27]	2020	J of the Endocrine Society	1 case	low	normal	low	high

In patients of reproductive age, when fertility preservation is desired, for early FIGO stages (IA) GCT, unilateral oophorectomy with multiple peritoneal biopsies, and washing cytology is acceptable. The complete surgical treatment is postponed after childbearing and completion of family planning [12,21]. In Stage IA postoperative treatment is not necessary, due to the good prognosis, adjuvant chemotherapy being offered in advanced stages. After surgery, if fertility was spared, the hormones levels return to normal. Long-term follow-up with clinical examination, US and inhibin B levels is recommended due to the late recurrence tendency of GCTs [21,28].

Granulosa cell tumors are hormonally active tumors, causing transient infertility as shown in our case. An associated risk factor for infertility was the proximal obstruction of the left tube while the right tube was normal with positive methylene-blue test during laparoscopy [21].

Currently there are no imaging criteria for the diagnostic of GCTs due to the rarity of these tumors [11]. Moreover, an anechoic ultrasonographic adnexal mass may be misdiagnosed as a benign finding and the right treatment can be delayed [11]. In the literature, the GCT is, usually, described as voluminous tumor (>10 cm diameter), but it was demonstrated that the dimension of the tumor can vary from a nonpalpable lesion to giant masses (between 3 and 24 cm) [3]. The particularity of our case was the diagnosis of a small size anechoic tumor of only 10 cm^ 3^ volume, during the infertility investigations in a patient with secondary amenorrhea. This anechoic mass of only 3 cm diameter could have been misdiagnosed even with a luteum corpus due to the increased peripheral vascularity.

Granulosa cell tumor of the ovary has a favorable prognosis and for patients of reproductive age fertility preservation is crucial. In order to establish imaging criteria for the diagnosis of GCTs, further research is needed.

In summary, endocrine disorders need the evaluation of inhibin B levels even if the ovaries appear normal on ultrasound examination, aiming not to omit an occult ovarian tumor. Elevated inhibin may be sign of the presence of an unsuspected tumor.

## Figures and Tables

**Figure 1 medicina-57-00829-f001:**
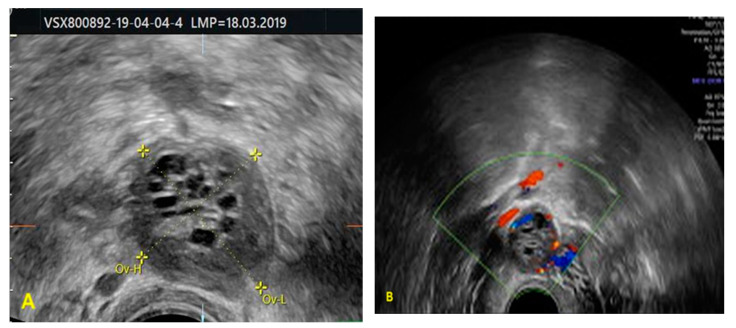
(**A**) Transvaginal ultrasonography—the right ovary (**B**) Transvaginal ultrasonography—increased peripheral vascularity of the right ovary.

**Figure 2 medicina-57-00829-f002:**
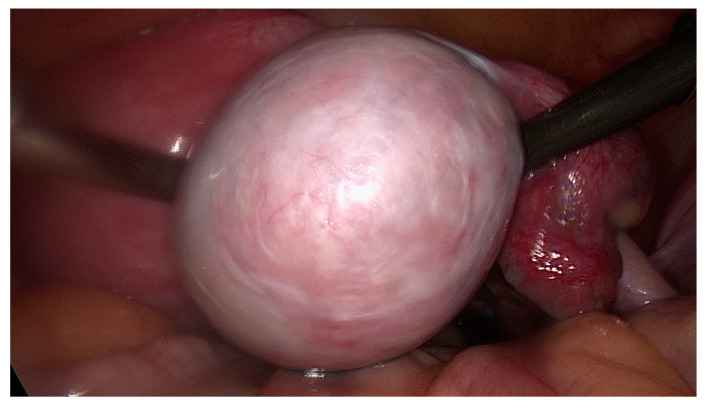
Laparoscopy—white enlarged right ovary, due to the presence of adult GCT.

**Figure 3 medicina-57-00829-f003:**
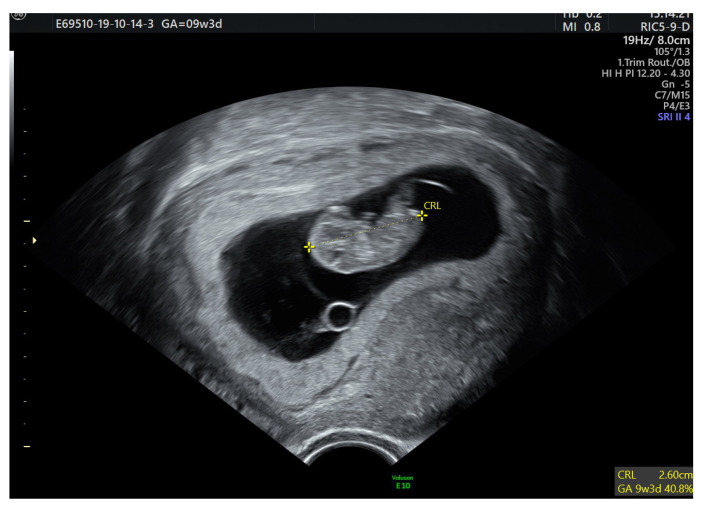
An early 26 mm embryo, corresponding to 9 weeks 3 days of amenorrhea.

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
