# Peer review of "Secondary Amenorrhea and Infertility Due to an Inhibin B Producing Granulosa Cell Tumor of the Ovary. A Rare Case Report and Literature Review"

_medicina, 2021, doi:10.3390/medicina57080829_

Round 1

Reviewer 1 Report

A nice and instructive case report and review of the literature.

this is a well written case report and narrative review. However, it contains nothing new and thus cannot be improved.  And language editing recommended.

Author Response

A nice and instructive case report and review of the literature.

Thank you for your appreciations.

This is a well written case report and narrative review. However, it contains nothing new and thus cannot be improved.  And language editing recommended.

Thank you for your recommendations: We performed English and punctuation edits.

Reviewer 2 Report

The article presents an important topic concerning the diagnosis of ovarian cancer, especially in young women. Granosa cell tumors (GCT) account for 70% of the genital tract-stromal tumors of the ovary, approximately 70%. There are two forms: adult GCT and juvenile GCT. The diagnosis of ovarian tumors in young women is particularly difficult. It is related to fertility preserving procedures.

A feature of these tumors is the ability to synthesize and secrete sex hormones. They consist of cells derived from sex lines that differentiate into female gonads (granular cells) or male (Sertoli cells) and / or female gonadal stromal cells (tecal cells) or male gonadal stromal cells (Leydig cells).

GCTs most often secrete the following hormones: AMH, Inhibin B, Inhibin A.

Tumor markers play an important role in the comprehensive diagnosis of patients with malignant neoplasms of the ovary. Their determination is not burdensome as other diagnostic methods and is of great importance in the differential diagnosis, in assessing the clinical progress of the disease and its prognosis. A significantly elevated level of tumor markers may indicate not only the presence of a tumor, but also that the disease is advanced.

This knowledge prior to surgery may increase the chances of a correct tumor resection.

However, in general real-world practice, hormonal tumor markers have been used primarily in treatment monitoring. 

Author Response

The article presents an important topic concerning the diagnosis of ovarian cancer, especially in young women. Granosa cell tumors (GCT) account for 70% of the genital tract-stromal tumors of the ovary, approximately 70%. There are two forms: adult GCT and juvenile GCT. The diagnosis of ovarian tumors in young women is particularly difficult. It is related to fertility preserving procedures. 

A feature of these tumors is the ability to synthesize and secrete sex hormones. They consist of cells derived from sex lines that differentiate into female gonads (granular cells) or male (Sertoli cells) and / or female gonadal stromal cells (tecal cells) or male gonadal stromal cells (Leydig cells). 

GCTs most often secrete the following hormones: AMH, Inhibin B, Inhibin A. 

Tumor markers play an important role in the comprehensive diagnosis of patients with malignant neoplasms of the ovary. Their determination is not burdensome as other diagnostic methods and is of great importance in the differential diagnosis, in assessing the clinical progress of the disease and its prognosis. A significantly elevated level of tumor markers may indicate not only the presence of a tumor, but also that the disease is advanced. 

This knowledge prior to surgery may increase the chances of a correct tumor resection. 

However, in general real-world practice, hormonal tumor markers have been used primarily in treatment monitoring. 

Thank you for your appreciations and for this informations.

Reviewer 3 Report

Good and interesting case report as Inhibin B is not routinely requested .

Data presented beautifully 

Author Response

Good and interesting case report as Inhibin B is not routinely requested .

Data presented beautifully 

Thank you for your appreciations.